# Domain Separation Networks

**Konstantinos Bousmalis**[*]
Google Brain
Mountain View, CA
konstantinos@google.com

**George Trigeorgis**[*][†]
Imperial College London
London, UK
g.trigeorgis@imperial.ac.uk

**Nathan Silberman**
Google Research
New York, NY
nsilberman@google.com

**Dilip Krishnan**
Google Research
Cambridge, MA
dilipkay@google.com

**Dumitru Erhan**
Google Brain
Mountain View, CA
dumitru@google.com

## Abstract

The cost of large scale data collection and annotation often makes the application of machine learning algorithms to new tasks or datasets prohibitively expensive. One approach circumventing this cost is training models on synthetic data where annotations are provided automatically. Despite their appeal, such models often fail to generalize from synthetic to real images, necessitating domain adaptation algorithms to manipulate these models before they can be successfully applied. Existing approaches focus either on mapping representations from one domain to the other, or on learning to extract features that are invariant to the domain from which they were extracted. However, by focusing only on creating a mapping or shared representation between the two domains, they ignore the individual characteristics of each domain. We hypothesize that explicitly modeling what is unique to each domain can improve a model's ability to extract domain-invariant features. Inspired by work on private-shared component analysis, we explicitly learn to extract image representations that are partitioned into two subspaces: one component which is private to each domain and one which is shared across domains. Our model is trained to not only perform the task we care about in the source domain, but also to use the partitioned representation to reconstruct the images from both domains. Our novel architecture results in a model that outperforms the state-of-the-art on a range of unsupervised domain adaptation scenarios and additionally produces visualizations of the private and shared representations enabling interpretation of the domain adaptation process.

## 1 Introduction

The recent success of supervised learning algorithms has been partially attributed to the large-scale datasets [16, 22] on which they are trained. Unfortunately, collecting, annotating, and curating such datasets is an extremely expensive and time-consuming process. An alternative would be creating large-scale datasets in non-realistic but inexpensive settings, such as computer generated scenes. While such approaches offer the promise of effectively unlimited amounts of labeled data, models trained in such settings do not generalize well to realistic domains. Motivated by this, we examine the problem of learning representations that are domain–invariant in scenarios where the data distributions during training and testing are different. In this setting, the source data is labeled for a particular task and we would like to transfer knowledge from the source to the target domain for which we have no ground truth labels.

In this work, we focus on the tasks of object classification and pose estimation, where the object of interest is in the foreground of a given image, for both source and target domains. The source and

---

[*]Authors contributed equally.
[†]This work was completed while George Trigeorgis was at Google Brain in Mountain View, CA.

target pixel distributions can differ in a number of ways. We define "low-level" differences in the distributions as those arising due to noise, resolution, illumination and color. "High-level" differences relate to the number of classes, the types of objects, and geometric variations, such as 3D position and pose. We assume that our source and target domains differ mainly in terms of the distribution of low level image statistics and that they have high level parameters with similar distributions and the same label space.

We propose a novel architecture, which we call Domain Separation Networks (DSN), to learn domain-invariant representations. Previous work attempts to either find a mapping from representations of the source domain to those of the target [26], or find representations that are shared between the two domains [8, 28, 17]. While this, in principle, is a good idea, it leaves the shared representations vulnerable to contamination by noise that is correlated with the underlying shared distribution [24]. Our model, in contrast, introduces the notion of a private subspace for each domain, which captures domain specific properties, such as background and low level image statistics. A shared subspace, enforced through the use of autoencoders and explicit loss functions, captures representations shared by the domains. By finding a shared subspace that is orthogonal to the subspaces that are private, our model is able to separate the information that is unique to each domain, and in the process produce representations that are more meaningful for the task at hand. Our method outperforms the state-of-the-art domain adaptation techniques on a range of datasets for object classification and pose estimation, while having an interpretability advantage by allowing the visualization of these private and shared representations. In Sec. 2, we survey related work and introduce relevant terminology. Our architecture, loss functions, and learning regime are presented in Sec. 3. Experimental results and discussion are given in Sec. 4. Finally, conclusions and directions for future work are in Sec. 5.

## 2   Related Work

Learning to perform unsupervised domain adaptation is an open theoretical and practical problem. While much prior art exists, our literature review focuses primarily on Convolutional Neural Network (CNN) based methods due to their empirical superiority on this problem [8, 17, 26, 29]. Ben-David et al. [4] provide upper bounds on a domain-adapted classifier in the target domain. They introduce the idea of training a binary classifier trained to distinguish source and target domains. The error that this "domain incoherence" classifier provides (along with the error of a source domain specific classifier) combine to give the overall bounds. Mansour et al. [18] extend the theory of [4] to handle the case of multiple source domains.

Ganin et al. [7, 8] and Ajakan et al. [2] use adversarial training to find domain–invariant representations in-network. Their Domain–Adversarial Neural Networks (DANN) exhibit an architecture whose first few feature extraction layers are shared by two classifiers trained simultaneously. The first is trained to correctly predict task-specific class labels on the source data while the second is trained to predict the domain of each input. DANN minimizes the domain classification loss with respect to parameters specific to the domain classifier, while maximizing it with respect to the parameters that are common to both classifiers. This minimax optimization becomes possible via the use of a gradient reversal layer (GRL).

Tzeng et al. [29] and Long et al. [17] proposed versions of this model where the maximization of the domain classification loss is replaced by the minimization of the Maximum Mean Discrepancy (MMD) metric [11]. The MMD metric is computed between features extracted from sets of samples from each domain. The Deep Domain Confusion Network by Tzeng et al. [29] has an MMD loss at one layer in the CNN architecture while Long et al. [17] proposed the Deep Adaptation Network that has MMD losses at multiple layers.

Other related techniques involve learning a transformation from one domain to the other. In this setup, the feature extraction pipeline is fixed during the domain adaptation optimization. This has been applied in various non-CNN based approaches [9, 5, 10] as well as the recent CNN-based Correlation Alignment (CORAL) [26] algorithm which "recolors" whitened source features with the covariance of features from the target domain.

## 3   Method

While the Domain Separation Networks (DSNs) could in principle be applicable to other learning tasks, without loss of generalization, we mainly use image classification as the cross-domain task. Given a labeled dataset in a source domain and an unlabeled dataset in a target domain, our goal is to train a classifier on data from the source domain that generalizes to the target domain. Like previous

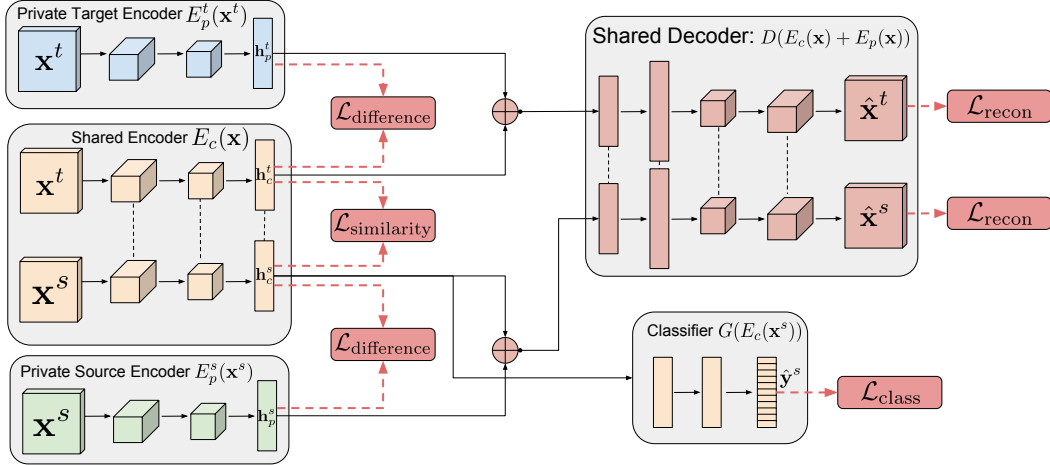

Figure 1: A shared-weight encoder $E_c(\mathbf{x})$ learns to capture representation components for a given input sample that are shared among domains. A private encoder $E_p(\mathbf{x})$ (one for each domain) learns to capture domain-specific components of the representation. A shared decoder learns to reconstruct the input sample by using both the private and source representations. The private and shared representation components are pushed apart with soft subspace orthogonality constraints $\mathcal{L}_{\text{difference}}$, whereas the shared representation components are kept similar with a similarity loss $\mathcal{L}_{\text{similarity}}$.

efforts [7, 8], our model is trained such that the representations of images from the source domain are similar to those from the target domain. This allows a classifier trained on images from the source domain to generalize as the inputs to the classifier are in theory invariant to the domain of origin. However, these representations might trivially include noise that is highly correlated with the shared representation, as shown by Salzmann et al. [24].

Our main novelty is that, inspired by recent work [14, 24, 30] on shared-space component analysis, DSNs explicitly model both private and shared components of the domain representations. The two private components of the representation are specific to each domain and the shared component of the representation is shared by both domains. To induce the model to produce such split representations, we add a loss function that encourages independence of these parts. Finally, to ensure that the private representations are still useful (avoiding trivial solutions) and to add generalizability, we also add a reconstruction loss. The combination of these objectives is a model that produces a shared representation that is similar for both domains and a private representation that is domain specific. By partitioning the space in such a manner, the classifier trained on the shared representation is better able to generalize across domains as its inputs are uncontaminated with aspects of the representation that are unique to each domain.

Let $\mathbf{X}_S = \{(\mathbf{x}_i^s, \mathbf{y}_i^s)\}_{i=0}^{N_s}$ represent a labeled dataset of $N_s$ samples from the source domain where $\mathbf{x}_i^s \sim \mathcal{D}_S$ and let $\mathbf{X}^t = \{\mathbf{x}_i^t\}_{i=0}^{N_t}$ represent an unlabeled dataset of $N_t$ samples from the target domain where $\mathbf{x}_i^t \sim \mathcal{D}_T$. Let $E_c(\mathbf{x}; \boldsymbol{\theta}_c)$ be a function parameterized by $\boldsymbol{\theta}_c$ which maps an image $\mathbf{x}$ to a hidden representation $\mathbf{h}_c$ representing features that are common or *shared* across domains. Let $E_p(\mathbf{x}; \boldsymbol{\theta}_p)$ be an analogous function which maps an image $\mathbf{x}$ to a hidden representation $\mathbf{h}_p$ representing features that are *private* to each domain. Let $D(\mathbf{h}; \boldsymbol{\theta}_d)$ be a decoding function mapping a hidden representation $\mathbf{h}$ to an image reconstruction $\hat{\mathbf{x}}$. Finally, $G(\mathbf{h}; \boldsymbol{\theta}_g)$ represents a task-specific function, parameterized by $\boldsymbol{\theta}_g$ that maps from hidden representations $\mathbf{h}$ to the task-specific predictions $\hat{\mathbf{y}}$. The resulting Domain Separation Network (DSN) model is depicted in Fig. 1.

## 3.1 Learning

Inference in a DSN model is given by $\hat{\mathbf{x}} = D(E_c(\mathbf{x}) + E_p(\mathbf{x}))$ and $\hat{\mathbf{y}} = G(E_c(\mathbf{x}))$ where $\hat{\mathbf{x}}$ is the reconstruction of the input $\mathbf{x}$ and $\hat{\mathbf{y}}$ is the task-specific prediction. The goal of training is to minimize the following loss with respect to parameters $\boldsymbol{\Theta} = \{\boldsymbol{\theta}_c, \boldsymbol{\theta}_p, \boldsymbol{\theta}_d, \boldsymbol{\theta}_g\}$:

$$\mathcal{L} = \mathcal{L}_{\text{task}} + \alpha\, \mathcal{L}_{\text{recon}} + \beta\, \mathcal{L}_{\text{difference}} + \gamma\, \mathcal{L}_{\text{similarity}} \tag{1}$$

where $\alpha, \beta, \gamma$ are weights that control the interaction of the loss terms. The classification loss $\mathcal{L}_{\text{task}}$ trains the model to predict the output labels we are ultimately interested in. Because we assume the target domain is unlabeled, the loss is applied only to the source domain. We want to minimize the negative log-likelihood of the ground truth class for each source domain sample:

$$\mathcal{L}_{\text{task}} = -\sum_{i=0}^{N_s} \mathbf{y}_i^s \cdot \log \hat{\mathbf{y}}_i^s, \tag{2}$$

where $\mathbf{y}_i^s$ is the one-hot encoding of the class label for source input $i$ and $\hat{\mathbf{y}}_i^s$ are the softmax predictions of the model: $\hat{\mathbf{y}}_i^s = G(E_c(\mathbf{x}_i^s))$. We use a scale-invariant mean squared error term [6] for the reconstruction loss $\mathcal{L}_{\text{recon}}$ which is applied to both domains:

$$\mathcal{L}_{\text{recon}} = \sum_{i=1}^{N_s} \mathcal{L}_{\text{si\_mse}}(\mathbf{x}_i^s, \hat{\mathbf{x}}_i^s) + \sum_{i=1}^{N_t} \mathcal{L}_{\text{si\_mse}}(\mathbf{x}_i^t, \hat{\mathbf{x}}_i^t) \tag{3}$$

$$\mathcal{L}_{\text{si\_mse}}(\mathbf{x}, \hat{\mathbf{x}}) = \frac{1}{k}\|\mathbf{x} - \hat{\mathbf{x}}\|_2^2 - \frac{1}{k^2}([\mathbf{x} - \hat{\mathbf{x}}] \cdot \mathbf{1}_k)^2, \tag{4}$$

where $k$ is the number of pixels in input $x$, $\mathbf{1}_k$ is a vector of ones of length $k$; and $\|\cdot\|_2^2$ is the squared $L_2$-norm. While a mean squared error loss is traditionally used for reconstruction tasks, it penalizes predictions that are correct up to a scaling term. Conversely, the scale-invariant mean squared error penalizes differences between *pairs* of pixels. This allows the model to learn to reproduce the overall shape of the objects being modeled without expending modeling power on the absolute color or intensity of the inputs. We validated that this reconstruction loss was indeed the correct choice experimentally in Sec. 4.3 by training a version of our best DSN model with the traditional mean squared error loss instead of the scale-invariant loss in Eq. 3.

The difference loss is also applied to both domains and encourages the shared and private encoders to encode different aspects of the inputs. We define the loss via a soft subspace orthogonality constraint between the private and shared representation of each domain. Let $\mathbf{H}_c^s$ and $\mathbf{H}_c^t$ be matrices whose rows are the hidden *shared* representations $\mathbf{h}_c^s = E_c(\mathbf{x}^s)$ and $\mathbf{h}_c^t = E_c(\mathbf{x}^t)$ from samples of source and target data respectively. Similarly, let $\mathbf{H}_p^s$ and $\mathbf{H}_p^t$ be matrices whose rows are the *private* representation $\mathbf{h}_p^s = E_p^s(\mathbf{x}^s)$ and $\mathbf{h}_p^t = E_p^t(\mathbf{x}^t)$ from samples of source and target data respectively[3]. The difference loss encourages orthogonality between the shared and the private representations:

$$\mathcal{L}_{\text{difference}} = \left\|\mathbf{H}_c^{s\top}\mathbf{H}_p^s\right\|_F^2 + \left\|\mathbf{H}_c^{t\top}\mathbf{H}_p^t\right\|_F^2, \tag{5}$$

where $\|\cdot\|_F^2$ is the squared Frobenius norm. Finally, $\mathcal{L}_{\text{similarity}}$ encourages the hidden representations $\mathbf{h}_c^s$ and $\mathbf{h}_c^t$ from the shared encoder to be as similar as possible irrespective of the domain. We experimented with two similarity losses, which we discuss in detail.

## 3.2 Similarity Losses

The domain adversarial similarity loss [7, 8] is used to train a model to produce representations such that a classifier cannot reliably predict the domain of the encoded representation. Maximizing such "confusion" is achieved via a Gradient Reversal Layer (GRL) and a *domain classifier* trained to predict the domain producing the hidden representation. The GRL has the same output as the identity function, but reverses the gradient direction. Formally, for some function $f(\mathbf{u})$, the GRL is defined as $Q\left(f(\mathbf{u})\right) = f(\mathbf{u})$ with a gradient $\frac{d}{d\mathbf{u}}Q(f(\mathbf{u})) = -\frac{d}{d\mathbf{u}}f(\mathbf{u})$. The domain classifier $Z(Q(\mathbf{h}_c); \boldsymbol{\theta}_z) \to \hat{d}$ parameterized by $\boldsymbol{\theta}_z$ maps a shared representation vector $\mathbf{h}_c = E_c(\mathbf{x}; \boldsymbol{\theta}_c)$ to a prediction of the label $\hat{d} \in \{0, 1\}$ of the input sample $\mathbf{x}$. Learning with a GRL is adversarial in that $\boldsymbol{\theta}_z$ is optimized to increase $Z$'s ability to discriminate between encodings of images from the source or target domains, while the reversal of the gradient results in the model parameters $\boldsymbol{\theta}_c$ learning representations from which domain classification accuracy is reduced. Essentially, we *maximize* the binomial cross-entropy for the domain prediction task with respect to $\boldsymbol{\theta}_z$, while *minimizing* it with respect to $\boldsymbol{\theta}_c$:

$$\mathcal{L}_{\text{similarity}}^{\text{DANN}} = \sum_{i=0}^{N_s+N_t} \left\{ d_i \log \hat{d}_i + (1 - d_i) \log(1 - \hat{d}_i) \right\}. \tag{6}$$

where $d_i \in \{0, 1\}$ is the ground truth domain label for sample $i$.

The Maximum Mean Discrepancy (MMD) loss [11] is a kernel-based distance function between pairs of samples. We use a biased statistic for the squared population MMD between shared encodings of the source samples $\mathbf{h}_c^s$ and the shared encodings of the target samples $\mathbf{h}_c^t$:

$$\mathcal{L}_{\text{similarity}}^{\text{MMD}} = \frac{1}{(N^s)^2} \sum_{i,j=0}^{N^s} \kappa(\mathbf{h}_{ci}^s, \mathbf{h}_{cj}^s) - \frac{2}{N^s N^t} \sum_{i,j=0}^{N^s, N^t} \kappa(\mathbf{h}_{ci}^s, \mathbf{h}_{cj}^t) + \frac{1}{(N^t)^2} \sum_{i,j=0}^{N^t} \kappa(\mathbf{h}_{ci}^t, \mathbf{h}_{cj}^t), \quad (7)$$

where $\kappa(\cdot, \cdot)$ is a PSD kernel function. In our experiments we used a linear combination of multiple RBF kernels: $\kappa(x_i, x_j) = \sum_n \eta_n \exp\{-\frac{1}{2\sigma_n}\|\mathbf{x}_i - \mathbf{x}_j\|^2\}$, where $\sigma_n$ is the standard deviation and $\eta_n$ is the weight for our $n^{th}$ RBF kernel. Any additional kernels we include in the multi–RBF kernel are additive and guarantee that their linear combination remains characteristic. Therefore, having a large range of kernels is beneficial since the distributions of the shared features change during learning, and different components of the multi–RBF kernel might be responsible at different times for making sure we reject a false null hypothesis, i.e. that the loss is sufficiently high when the distributions are not similar [17]. The advantage of using an RBF kernel with the MMD distance is that the Taylor expansion of the Gaussian function allows us to match all the moments of the two populations. The caveat is that it requires finding optimal kernel bandwidths $\sigma_n$.

## 4 Evaluation

We are motivated by the problem of learning models on a clean, synthetic dataset and testing on noisy, real–world dataset. To this end, we evaluate on object classification datasets used in previous work[4] including MNIST and MNIST-M [8], the German Traffic Signs Recognition Benchmark (GTSRB) [25], and the Streetview House Numbers (SVHN) [20]. We also evaluate on the cropped LINEMOD dataset, a standard for object instance recognition and 3D pose estimation [12, 31], for which we have synthetic and real data[5]. We tested the following unsupervised domain adaptation scenarios: *(a)* from MNIST to MNIST-M; *(b)* from SVHN to MNIST; *(c)* from synthetic traffic signs to real ones with GTSRB; *(d)* from synthetic LINEMOD object instances rendered on a black background to the same object instances in the real world.

We evaluate the efficacy of our method with each of the two similarity losses outlined in Sec. 3.2 by comparing against the prevailing visual domain adaptation techniques for neural networks: Correlation Alignment (CORAL) [26], Domain-Adversarial Neural Networks (DANN) [7, 8], and MMD regularization [29, 17]. For each scenario we provide two additional baselines: the performance on the target domain of the respective model with no domain adaptation and trained *(a)* on the source domain ("Source-only" in Tab. 1) and *(b)* on the target domain ("Target-only"), as an empirical lower and upper bound respectively.

We have not found a universally applicable way to optimize hyperparameters for unsupervised domain adaptation. Previous work [8] suggests the use of reverse validation. We implemented this (see Supplementary Material for details) but found that that the reverse validation accuracy did not always align well with test accuracy. Ideally we would like to avoid using labels from the target domain, as it can be argued that if ones does have target domain labels, they should be used during training. However, there are applications where a labeled target domain set cannot be used for training. An example is the labeling of a dataset with the use of AprilTags [21], 2D barcodes that can be used to label the pose of an object, provided that a camera is calibrated and the physical dimensions of the barcode are known. These images should not be used when learning features from pixels, because the model might be able to decipher the tags. However, they can be part of a test set that is not available during training, and an equivalent dataset without the tags could be used for unsupervised domain adaptation. We thus chose to use a small set of labeled target domain data as a validation set for

Table 1: Mean classification accuracy (%) for the unsupervised domain adaptation scenarios we evaluated all the methods on. We have replicated the experiments from Ganin et al. [8] and in parentheses we show the results reported in their paper. The "Source-only" and "Target-only" rows are the results on the target domain when using no domain adaptation and training only on the source or the target domain respectively.

| Model | MNIST to MNIST-M | Synth Digits to SVHN | SVHN to MNIST | Synth Signs to GTSRB |
|---|---|---|---|---|
| Source-only | 56.6 (52.2) | 86.7 (86.7) | 59.2 (54.9) | 85.1 (79.0) |
| CORAL [26] | 57.7 | 85.2 | 63.1 | 86.9 |
| MMD [29, 17] | 76.9 | 88.0 | 71.1 | 91.1 |
| DANN [8] | 77.4 (76.6) | 90.3 (91.0) | 70.7 (73.8) | 92.9 (88.6) |
| DSN w/ MMD (ours) | 80.5 | 88.5 | 72.2 | 92.6 |
| DSN w/ DANN (ours) | **83.2** | **91.2** | **82.7** | **93.1** |
| Target-only | 98.7 | 92.4 | 99.5 | 99.8 |

the hyperparameters of all the methods we compare. All methods were evaluated using the same protocol, so comparison numbers are fair and meaningful. The performance on this validation set can serve as an *upper bound* of a satisfactory validation metric for unsupervised domain adaptation, which to our knowledge validating the parameters in an unsupervised manner is still an open research question, and out of the scope of this work.

## 4.1 Datasets and Adaptation Scenarios

**MNIST to MNIST-M.** In this domain adaptation scenario we use the popular MNIST [15] dataset of handwritten digits as the source domain, and MNIST-M, a variation of MNIST proposed for unsupervised domain adaptation by [8]. MNIST-M was created by using each MNIST digit as a binary mask and inverting with it the colors of a background image. The background images are random crops uniformly sampled from the Berkeley Segmentation Data Set (BSDS500) [3]. In all our experiments, following the experimental protocol by [8]. Out of the $59,001$ MNIST-M training examples, we used the labels for $1,000$ of them to find optimal hyperparameters for our models. This scenario, like all three digit adaptation scenarios, has 10 class labels.

**Synthetic Digits to SVHN.** In this scenario we aim to learn a classifier for the Street-View House Number data set (SVHN) [20], our target domain, from a dataset of purely synthesized digits, our source domain. The synthetic digits [8] dataset was created by rasterizing bitmap fonts in a sequence (one, two, and three digits) with the ground truth label being the digit in the center of the image, just like in SVHN. The source domain samples are further augmented by variations in scale, translation, background colors, stroke colors, and Gaussian blurring. We use $479,400$ Synthetic Digits for our source domain training set, $73,257$ unlabeled SVHN samples for domain adaptation, and $26,032$ SVHN samples for testing. Similarly to above, we use the labels of $1,000$ SVHN training examples for hyperparameter validation.

**SVHN to MNIST.** Although the SVHN dataset contains significant variations (in scale, background clutter, blurring, embossing, slanting, contrast, rotation, sequences to name a few) there is not a lot of variation in the actual digits shapes. This makes it quite distinct from a dataset of handwritten digits, like MNIST, where there are a lot of elastic distortions in the shapes, variations in thickness, and noise on the digits themselves. Since the ground truth digits in both datasets are centered, this is a well-posed and rather difficult domain adaptation scenario. As above, we used the labels of $1,000$ MNIST training examples for validation.

**Synthetic Signs to GTSRB.** We also perform an experiment using a dataset of synthetic traffic signs from [19] to real world dataset of traffic signs (GTSRB) [25]. While the three-digit adaptation scenarios have 10 class labels, this scenario has 43 different traffic signs. The synthetic signs were obtained by taking relevant pictograms and adding various types of variations, including random backgrounds, brightness, saturation, 3D rotations, Gaussian and motion blur. We use $90,000$ synthetic signs for training, $1,280$ random GTSRB real-world signs for domain adaptation and validation, and the remaining $37,929$ GTSRB real signs as the test set.

Table 2: Mean classification accuracy and pose error for the "Synth Objects to LINEMOD" scenario.

| Method | Classification Accuracy | Mean Angle Error |
|---|---|---|
| Source-only | 47.33% | 89.2° |
| MMD | 72.35% | 70.62° |
| DANN | 99.90% | 56.58° |
| DSN w/ MMD (ours) | 99.72% | 66.49° |
| DSN w/ DANN (ours) | **100.00%** | **53.27°** |
| Target-only | 100.00% | 6.47° |

**Synthetic Objects to LineMod.** The LineMod dataset [31] consists of CAD models of objects in a cluttered environment and a high variance of 3D poses for each object. We use the 11 non-symmetric objects from the cropped version of the dataset, where the images are cropped with the object in the center, for the task of object instance recognition and 3D pose estimation. We train our models on $16,962$ images for these objects rendered on a black background without additional noise. We use a target domain training set of $10,673$ real-world images for domain adaptation and validation, and a target domain test set of $2,655$ for testing. For this scenario our task is both classification and pose estimation; our task loss is therefore $\mathcal{L}_{\text{task}} = \sum_{i=0}^{N_s}\{-\mathbf{y}_i^s \cdot \log \hat{\mathbf{y}}_i^s + \xi \log(1 - |\mathbf{q}^s \cdot \hat{\mathbf{q}}^s|)\}$, where $\mathbf{q}^s$ is the positive unit quaternion vector representing the ground truth 3D pose, and $\hat{\mathbf{q}}^s$ is the equivalent prediction. The first term is the classification loss, similar to the rest of the experiments, the second term is the log of a 3D rotation metric for quaternions [13], and $\xi$ is the weight for the pose loss. In Tab. 2 we report the mean angle the object would need to be rotated (on a fixed 3D axis) to move from the predicted to the ground truth pose [12].

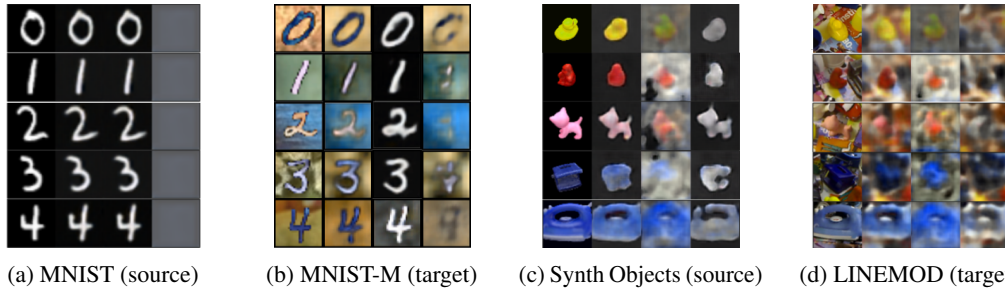

(a) MNIST (source)    (b) MNIST-M (target)    (c) Synth Objects (source)    (d) LINEMOD (target)

Figure 2: Reconstructions for the representations of the two domains for "MNIST to MNIST-M" and for "Synth Objects to LINEMOD". In each block from left to right: the original image $\mathbf{x}_t$; reconstructed image $D(E_c(\mathbf{x}^t) + E_p(\mathbf{x}^t))$; shared only reconstruction $D(E_c(\mathbf{x}^t))$; private only reconstruction $D(E_p(\mathbf{x}^t))$.

## 4.2 Implementation Details

All the models were implemented using TensorFlow [6] [1] and were trained with Stochastic Gradient Descent plus momentum [27]. Our initial learning rate was multiplied by 0.9 every $20,000$ steps (mini-batches). We used batches of 32 samples from each domain for a total of 64 and the input images were mean-centered and rescaled to $[-1, 1]$. In order to avoid distractions for the main classification task during the early stages of the training procedure, we activate any additional domain adaptation loss after $10,000$ steps of training. For all our experiments our CNN topologies are based on the ones used in [8], to be comparable to previous work in unsupervised domain adaptation. The exact architectures for all models are shown in our Supplementary Material.

In our framework, CORAL [26] would be equivalent to fixing our shared representation matrices $\mathbf{H}_c^s$ and $\mathbf{H}_c^t$, normalizing them and then minimizing $\|\mathbf{A}\mathbf{H}_c^{s\top}\mathbf{H}_c^s\mathbf{A}^\top - \mathbf{H}_c^{t\top}\mathbf{H}_c^t\|_F^2$ with respect to a weight matrix $\mathbf{A}$ that aligns the two correlation matrices. For the CORAL experiments, we follow the suggestions of [26], and extract features for both source and target domains from the penultimate layer of each network. Once the correlation matrices for each domain are aligned, we evaluate on

Table 3: Effect of our difference and reconstruction losses on our best model. The first row is replicated from Tab. 1. In the second row, we remove the soft orthogonality constraint. In the third row, we replace the scale-invariant MSE with regular MSE.

| Model | MNIST to MNIST-M | Synth. Digits to SVHN | SVHN to MNIST | Synth. Signs to GTSRB |
|-------|------------------|------------------------|---------------|------------------------|
| All terms | **83.23** | **91.22** | **82.78** | **93.01** |
| No $\mathcal{L}_{\text{difference}}$ | 80.26 | 89.21 | 80.54 | 91.89 |
| With $\mathcal{L}_{\text{recon}}^{L2}$ | 80.42 | 88.98 | 79.45 | 92.11 |

the target test data the performance of a linear support vector machine (SVM) classifier trained on the source training data. The SVM penalty parameter was optimized based on the target domain validation set for each of our domain adaptation scenarios. For MMD regularization, we used a linear combination of 19 RBF kernels (details can be found in the Supplementary Material). Preliminary experiments with having MMD applied on more than one layers did not show any performance improvement for our experiments and architectures. For DANN regularization, we applied the GRL and the domain classifier as prescribed in [8] for each scenario.

For our Domain Separation Network experiments, our similarity losses are always applied at the first fully connected layer of each network after a number of convolutional and max pooling layers. For each private space encoder network we use a simple convolutional and max pooling structure followed by a fully-connected layer with a number of nodes equal to the number of nodes at the final layer $\mathbf{h}_c$ of the equivalent shared encoder $E_c$. The output of the shared and private encoders gets added before being fed to the shared decoder $D$.

## 4.3 Discussion

The DSN with DANN model outperforms all the other methods we experimented with for all our unsupervised domain adaptation scenarios (see Tab. 1 and 2). Our unsupervised domain separation networks are able to improve both upon MMD regularization and DANN. Using DANN as a similarity loss (Eq. 6) worked better than using MMD (Eq. 7) as a similarity loss, which is consistent with results obtained for domain adaptation using MMD regularization and DANN alone.

In order to examine the effect of the soft orthogonality constraints ($\mathcal{L}_{\text{difference}}$), we took our best model, our DSN model with the DANN loss, and removed these constraints by setting the $\beta$ coefficient to 0. Without them, the model performed consistently worse in all scenarios. We also validated our choice of our scale-invariant mean squared error reconstruction loss as opposed to the more popular mean squared error loss by running our best model with $\mathcal{L}_{\text{recon}}^{L2} = \frac{1}{k}||\mathbf{x} - \hat{\mathbf{x}}||_2^2$. With this variation we also get worse classification results consistently, as shown in experiments from Tab. 3.

The shared and private representations of each domain are combined for the reconstruction of samples. Individually decoding the shared and private representations gives us reconstructions that serve as useful depictions of our domain adaptation process. In Fig. 2 we use the "MNIST to MNIST-M" and the "Synth. Objects to LINEMOD" scenarios for such visualizations. In the former scenario, the model cleanly separates the foreground from the background and produces a shared space that is very similar to the source domain. This is expected since the target is a transformation of the source. In the latter scenario, the model is able to produce visualizations of the shared representation that look very similar between source and target domains, which are useful for classification and pose estimation.

## 5 Conclusion

We present in this work a deep learning model that improves upon existing unsupervised domain adaptation techniques. The model does so by explicitly separating representations private to each domain and shared between source and target domains. By using existing domain adaptation techniques to make the shared representations similar, and soft subspace orthogonality constraints to make private and shared representations dissimilar, our method outperforms all existing unsupervised domain adaptation methods in a number of adaptation scenarios that focus on the synthetic-to-real paradigm.

**Acknowledgments**

We would like to thank Samy Bengio, Kevin Murphy, and Vincent Vanhoucke for valuable comments on this work. We would also like to thank Yaroslav Ganin and Paul Wohlhart for providing some of the datasets we used.

## Footnotes

[3]The matrices are transformed to have zero mean and unit $l_2$ norm.

[4]The most commonly used dataset for visual domain adaptation in the context of object classification is Office [23]. However, this dataset exhibits significant variations in both low-level and high-level parameter distributions. Low-level variations are due to the different cameras and background textures in the images (e.g. Amazon versus DSLR). However, there are significant high-level variations due to object identity: e.g. the motorcycle class contains non-motorcycle objects; the backpack class contains a laptop; some domains contain the object in only one pose. Other commonly used datasets such as Caltech-256 suffer from similar problems. We therefore exclude these datasets from our evaluation. For more information, see our Supplementary Material.

[5]https://cvarlab.icg.tugraz.at/projects/3d_object_detection/

[6] We provide code at `https://github.com/tensorflow/models/domain_adaptation`.

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
