[Supplementary Material]

# Supplementary of "Domain Separation Networks"

## 1  Office Dataset Criticism

The most commonly used dataset for visual domain adaptation in the context of object classification is Office [4], sometimes combined with the Caltech–256 dataset [2] as an additional domain. However, these datasets exhibit significant variations in both low-level and high-level parameter distributions. Low-level variations are due to the different cameras and background textures in the images (e.g. Amazon versus DSLR), which is welcome. However, there are significant high-level variations due to elements like label pollution: e.g. the motorcycle class contains non-motorcycle objects; the backpack class contains 2 laptops; some classes contain the object in only one pose. Other commonly used datasets such as Caltech-256 suffer from similar problems. We illustrate some of these issues for the 'back_pack' class for its 92 Amazon samples, its 12 DSLR samples, its 29 Webcam samples, and its 151 Caltech samples in Fig. 1. Other classes exhibit similar problems. For these reasons some works, eg [5], pretrain their models on Imagenet before performing the domain adaptation in these scenarios. This essentially involves another source domain (Imagenet) in the transfer.

Figure 1: Examples of the 'back_pack' class in the different domains in Office and Caltech–256. **First Row:** 5 of the 92 images in the Amazon domain. **Second Row:** The DSLR domain contains 4 images for the rightmost image from different frontal angles, 2 images for the other 4 backpacks for a total of 12 images for this class. **Third Row:** The webcam domain contains the exact same backpacks with DSLR with similar poses for a total of 29 images for this class. **Fourth Row:** Some of the 151 backpack samples Caltech domain.

## 2 Correlation Regularization

Correlation Alignment (CORAL) [5] aims to find a mapping from the representations of the source domain to the representations of the target domain by matching only the second–order statistics. In our framework, this would be equivalent to fixing our common representation matrices $\mathbf{H}_c^s$ and $\mathbf{H}_c^t$ after normalizing them and then finding a weight matrix $\hat{\mathbf{A}} = \underset{\mathbf{A}}{\operatorname{argmin}} \left\| \mathbf{A}{\mathbf{H}_c^s}^{\top}\mathbf{H}_c^s\mathbf{A}^{\top} - {\mathbf{H}_c^t}^{\top}\mathbf{H}_c^t \right\|_F^2$ that aligns the two correlation matrices. Although this has the advantage that the optimization is convex and can be solved in closed form, all convolutional features remain fixed during the process, which might not be optimal for the task at hand. Also, because of this we are not able to use it as a similarity loss for our DSNs. Motivated by this shortcoming, we propose here a new domain adaptation method, Correlation Regularization (CorReg). We show in Tab. 1 that our new domain adaptation method, which is theoretically as powerful as an MMD loss with a second–order polynomial kernel, outperforms CORAL in all our datasets. Adapting a feature hierarchy to be domain–invariant is more powerful than learning a mapping from the representations of one domain to those of another. Moreover, we use it as yet another similarity loss for our Domain Separation Networks:

$$\mathcal{L}_{\text{similarity}}^{\text{CorReg}} = \left\| {\mathbf{H}_c^s}^{\top}\mathbf{H}_c^s - {\mathbf{H}_c^t}^{\top}\mathbf{H}_c^t \right\|_F^2 \tag{1}$$

Our DNS with CorReg performs better than both CORAL and CorReg, which is consistent with the rest of our results.

Table 1: Our main results from the paper with two additional lines for CorReg and DSN with CorReg.

| Model | MNIST to MNIST-M | Synth Digits to SVHN | SVHN to MNIST | Synth Signs to GTSRB |
|---|---|---|---|---|
| Source-only | 56.6 (52.2) | 86.7 (86.7) | 59.2 (54.9) | 85.1 (79.0) |
| CORAL [5] | 57.7 | 85.2 | 63.1 | 86.9 |
| CorReg (Ours) | 62.06 | 87.33 | 69.20 | 90.75 |
| MMD [6, 3] | 76.9 | 88.0 | 71.1 | 91.1 |
| DANN [1] | 77.4 (76.6) | 90.3 (91.0) | 70.7 (73.8) | 92.9 (88.6) |
| DSN w/ MMD (ours) | 80.5 | 88.5 | 72.2 | 92.6 |
| DSN w/ DANN (ours) | **83.2** | **91.2** | **82.7** | **93.1** |
| Target-only | 98.7 | 92.4 | 99.5 | 99.8 |

## 3 Network Topologies and Optimal Parameters

Since we used different network topologies for our domain adaptation scenarios, there was not enough space to include these in the main paper. We present the exact topologies used in Figures 2–5.

Similarly, we list here all hyperparameters that are important for total reproducibility of all our results. For CORAL, the SVM penalty parameter that was optimized based on the validation set for each of our domain adaptation scenarios: $1e^{-4}$ for "MNIST to MNIST-M", "Synth Digits to SVHN", "Synth Signs to GTSRB", and $1e^{-3}$ for "SVHN to MNIST". For MMD we use 19 RBF kernels with the following standard deviation parameters:

$$\boldsymbol{\sigma} = [10^{-6}, 10^{-5}, 10^{-4}, 10^{-3}, 10^{-2}, 10^{-1}, 1, 5, 10, 15, 20, 25, 30, 35, 100, 10^3, 10^4, 10^5, 10^6]$$

and equal $\eta$ weights. We use learning rate between $[0.01, 0.015]$ and $\gamma \in [0.1, 0.3]$. For DANN we use learning rate between $[0.01, 0.015]$ and $\gamma \in [0.15, 0.25]$. For DSN w/ DANN and DSN w/ MMD we use a constant initial learning rate of $0.01$ use the hyperparameters in the range of: $\alpha \in [0.01, 0.15], \beta \in [0.05, 0.075], \gamma \in [0.25, 0.3]$, whereas for DNS w/ CorReg we use $\gamma \in [20, 100]$. For the GTSRB experiment we use $\alpha \in [0.01, 0.015]$. In all cases we use an exponential decay of $0.95$ on the learning rate every $20,000$ iterations. For the LINEMOD experiments we use $\xi = 0.125$.

**shared** encoder $E_c(\cdot; \vartheta_c)$

| conv 5x5x32 ReLU | max-pool 2x2 2x2 stride | conv 5x5x48 ReLU | max-pool 2x2 2x2 stride | FC 100 units ReLU |
|---|---|---|---|---|

**private target** encoder $E_p^t(\cdot; \vartheta_p^t)$

| conv 5x5x32 ReLU | max-pool 2x2 2x2 stride | conv 5x5x64 ReLU | max-pool 2x2 2x2 stride | FC 100 units ReLU |
|---|---|---|---|---|

**private source** encoder $E_p^s(\cdot; \vartheta_p^s)$

| conv 5x5x32 ReLU | max-pool 2x2 2x2 stride | conv 5x5x64 ReLU | max-pool 2x2 2x2 stride | FC 100 units ReLU |
|---|---|---|---|---|

**shared decoder** $D(\cdot; \vartheta_d)$

| FC 300 units ReLU | reshape 10x10x3 | conv 5x5x16 ReLU | conv 5x5x16 ReLU | upsampling 32x32x16 | conv 3x3x16 ReLU | conv 3x3x3 |
|---|---|---|---|---|---|---|

**classifier** $G(\cdot; \vartheta_c)$

| FC 100 units ReLU | FC 10 units softmax |
|---|---|

**domain adversarial network** $Z(\cdot; \vartheta_z)$

| gradient reversal layer | FC 100 units ReLU | FC 1 unit |
|---|---|---|

Figure 2: The network topology for "MNIST to MNIST-M". The green blocks denote convolutional layers, the red pooling layers, the blue fully connected ones, and the yellow upsampling ones.

**shared** encoder $E_c(\cdot; \vartheta_c)$

| conv 5x5x64 ReLU | max-pool 3x3 2x2 stride | conv 5x5x64 ReLU | max-pool 3x3 2x2 stride | FC 3072 units ReLU |
|---|---|---|---|---|

**private target** encoder $E_p^t(\cdot; \vartheta_p^t)$

| conv 5x5x32 ReLU | max-pool 2x2 2x2 stride | conv 5x5x64 ReLU | max-pool 2x2 2x2 stride | FC 3072 units ReLU |
|---|---|---|---|---|

**private source** encoder $E_p^s(\cdot; \vartheta_p^s)$

| conv 5x5x32 ReLU | max-pool 2x2 2x2 stride | conv 5x5x64 ReLU | max-pool 2x2 2x2 stride | FC 3072 units ReLU |
|---|---|---|---|---|

**shared decoder** $D(\cdot; \vartheta_d)$

| FC 300 units ReLU | reshape 10x10x3 | conv 5x5x16 ReLU | conv 5x5x16 ReLU | upsampling 32x32x16 | conv 3x3x16 ReLU | conv 3x3x3 |
|---|---|---|---|---|---|---|

**classifier** $G(\cdot; \vartheta_c)$

| FC 2048 units ReLU | FC 10 units softmax |
|---|---|

**domain adversarial network** $Z(\cdot; \vartheta_z)$

| gradient reversal layer | FC 100 units ReLU | FC 1 unit |
|---|---|---|

Figure 3: The network topology for the "Synth SVHN to SVHN" and "SVHN to MNIST" experiments. The green blocks denote convolutional layers, the red pooling layers, the blue fully connected ones, and the yellow upsampling ones.

**shared** encoder $E_c(\cdot; \vartheta_c)$

| conv 5x5x96 ReLU | max-pool 2x2 2x2 stride | conv 3x3x144 ReLU | max-pool 2x2 2x2 stride | conv 5x5x256 ReLU | max-pool 2x2 2x2 stride |

**private target** encoder $E_p^t(\cdot; \vartheta_p^t)$

| conv 5x5x96 ReLU | max-pool 2x2 2x2 stride | conv 3x3x144 ReLU | max-pool 2x2 2x2 stride | conv 5x5x256 ReLU | max-pool 2x2 2x2 stride |

**private source** encoder $E_p^s(\cdot; \vartheta_p^s)$

| conv 5x5x96 ReLU | max-pool 2x2 2x2 stride | conv 5x5x144 ReLU | max-pool 2x2 2x2 stride | conv 5x5x256 ReLU | max-pool 2x2 2x2 stride |

**shared decoder** $D(\cdot; \vartheta_d)$

| conv 3x3x32 ReLU | upsampling 20x20x32 | conv 3x3x32 ReLU | upsampling 40x40x32 | conv 3x3x16 ReLU | conv 3x3x3 |

**classifier** $G(\cdot; \vartheta_c)$

| FC 512 units ReLU | FC 43 units softmax |

**domain adversarial network** $Z(\cdot; \vartheta_z)$

| gradient reversal layer | FC 100 units ReLU | FC 1 unit |

Figure 4: The network topology for "Synth Signs to GTSRB". The green blocks denote convolutional layers, the red pooling layers, the blue fully connected ones, and the yellow upsampling ones.

**shared** encoder $E_c(\cdot; \vartheta_c)$

| conv 5x5x32 ReLU | max-pool 2x2 2x2 stride | conv 3x3x64 ReLU | max-pool 2x2 2x2 stride | FC 128 units ReLU |

**private target** encoder $E_p^t(\cdot; \vartheta_p^t)$

| conv 5x5x32 ReLU | max-pool 2x2 2x2 stride | conv 5x5x64 ReLU | max-pool 2x2 2x2 stride | FC 128 units ReLU |

**private source** encoder $E_p^s(\cdot; \vartheta_p^s)$

| conv 5x5x32 ReLU | max-pool 2x2 2x2 stride | conv 5x5x64 ReLU | max-pool 2x2 2x2 stride | FC 128 units ReLU |

**shared decoder** $D(\cdot; \vartheta_d)$

| FC 600 units ReLU | conv 5x5x32 ReLU | upsampling 16x16x32 | conv 5x5x32 ReLU | upsampling 32x32x32 | conv 5x5x32 ReLU | upsampling 64x54x32 | conv 3x3x4 |

**task specific network** $G(\cdot; \vartheta_c)$

| FC 512 units ReLU | FC 11 units softmax / FC 4 units / L2 Normalization |

**domain adversarial network** $Z(\cdot; \vartheta_z)$

| gradient reversal layer | FC 100 units ReLU | FC 1 unit |

Figure 5: The network topology for "Synthetic Objects to Linemod". The green blocks denote convolutional layers, the red pooling layers, the blue fully connected ones, and the yellow upsampling ones.