[Reviews · NeurIPS 2016]

Reviewer 1

Summary

The paper describes a method for unsupervised domain adaptation applied to image classification tasks. It extends the recently proposed adversarial method implemented through a gradient reversal layer (Ganin and Lempitsky, 2015, ref [8]) and adds additional modeling which separates the feature representation into a shared space and a private space. Orthogonality is favored between shared and private spaces, and additional regularization is provided through reconstruction loss. The method is evaluated on 5 different scenarios and compared with the state of the art.

Qualitative Assessment

Strong points: The method builds on recent but successful reference [8] and extends it relaxing the previous constraints of complete feature invariance between source and domain. Private sub spaces w/o invariance allow the network to model properties of the data, which are not shared. While the separation into private and shared subspaces has been reported before, I am convinced by the originality of the approach, which combines this separation with the gradient reversal in a way, which makes sense. I also like the unsupervised regularization of the private mapping, which adds stability to the method. Evaluation is a strong point, with experiments on 5 different datasets and large gains reported with respect to the baselines, including [8]. Weaknesses : One could eventually object that adversarial domain adaptation is not new, and neither are projections into shared and private spaces and orthogonality constraints. However, these are minor points. I still think that the whole package is sufficiently novel even for a high level conference as NIPS. I am also wondering where the exact contribution of the private space actually comes from. The training loss related to the task classifier is unlikely to give any higher performance on the target data (by construction due to the orthogonality constraints). Minor remarks: - In equation (5), I think the loss should be HH^T and not H^T H if orthogonality is supposed to be favored and features are rows. - the task loss is called L_task in the text but L_class in figure 1

Confidence in this Review

3-Expert (read the paper in detail, know the area, quite certain of my opinion)


Reviewer 2

Summary

This paper builds on recent developments in deep learning community regarding domain adaptation in vision problems. In particular, the notion of private and shared representation is used to develop a new architecture for learning domain-invariant representations.

Qualitative Assessment

This a well-written paper on unsupervised domain adaptation building on the DANN and MMD objective functions and using the private and shared representation concept. While comparison with DANN and MMD techniques is ideal and meaningful, was there any comparison performed with some of the supervised techniques to understand the performance limit? Also, is there an indicator to suggest the need for domain adaptation?

Confidence in this Review

2-Confident (read it all; understood it all reasonably well)


Reviewer 3

Summary

This paper proposes to learn both common and domain-specific representations for domain adaptation from source domain to target domain. Experiments on digits and traffic sign dataset show good performance (even though not convincing even fatal flawed, see below).

Qualitative Assessment

------Initial Review----- This paper proposes Domain Separation Networks that learn both common and domain-specific representations for domain adaptation. I like the idea of domain separation even though it is not new. However, I do not think the current draft sufficiently validate the proposed approach. First of all, neither the experimental settings nor the datasets used are standard. To have a direct comparison to other methods (e.g., [7], [29], [17], [26], etc.), standard settings and benchmark datasets should be used. For the current draft, it is not clear the better performance is due to better hyper-parameter tuning on the validation set or from the proposed model. What’s more, since the baseline methods in the original papers are NOT tested on these datasets, it might be that with proper tuning of parameters on the validation set (that not available in the standard setting), the baseline methods achieve better performance. Thus, I encourage the authors to experiment on standard benchmark datasets with the standard protocol. Secondly, I do not think pixel-wise reconstruction is widely applicable. It might work for small gray-level digits data, but the space of real images is much larger (e.g., 256^5 for 256*256 RGB image). What’s more, pixel-level reconstruction prohibits invariance of features. Higher level reconstruction (e.g., conv3) might be better. Last but not least, as the model is very complicated, I would expect the author to provide sensitivity analysis of parameters/hyper-parameters and both mean and std of the accuracy by running the experiments multiple times. ------Additional Review after authors' feedback------ After reading all the reviews and authors' feedback, I think the CURRENT draft is not suitable for publication at NIPS. My concerns are for both the approach and the evaluation (experimental setting is flawed). I REALLY hope the authors would have taken the advice of three reviewers and test it on some real images of 3D objects. Since the authors did not provide any meaningful feedback (at least to my three points), I just have to delve deeper into the approach. Even though the novelty is limited, I liked the combination before. However, further reading confirms my opinion that the current draft has not sufficiently validated the approach. My personal conclusion now is that the proposed approach is NOT going to work on read image of 3D objects with DEEP networks. For classification of the digits/traffic signs, it is a pretty much solved problem and the experiment setting is also flawed. For experiment settings, I think it is flawed. All the baseline methods are designed to be purely UNSUPERVISED, NO labelled TARGET data is used anyway (training or validation). In this paper's setting, there are >1000 labeled TARGET data used for VALIDATION. As showed in recently published methods (e.g., ladder networks, virtual adversarial, etc.), 1000 (10/category) labeled images for digits are enough to train a strong classifier, might even stronger than the number reported in this paper with the more complicated adaptation method. I really can NOT see any point of using that large portion of labelled TARGET data to tune parameter for some complicated methods rather than train a much stronger and simpler classifier on it directly. Thus, in the 5 toy experiments designed by the authors, no adaptation (e.g., ladder networks, virtual adversarial, etc.) might achieve better performance than the methods proposed by the authors with adaptation. What's the point of use it as a validation set rather than train a classifier on it? A not very accurate analogy would be putting a gas powered generator on an electrical car to generate electricity for the car. Why can not I buy a car powered by gas directly? It is more energy efficient. For the authors' feedback, some of the issues pointed by fellow reviewers are very easy to address. However, the author just provided vague discussion instead of quantitative analysis/results. For example, the mean and std can be provided by running the same experiments many times. For the sensitivity test, the authors claim to be less sensitive than DANN. What's do you mean by less? How about comparisons to other methods mentioned in the paper? For the datasets, as mentioned by fellow reviewers, the 5 datasets are rather similar and constrained in several cases. The variance of digits or traffic signs is MUCH lower than real images. They are both 2D objects with pretty much the same viewpoints. For real images of 3D objects, due to change of pose and viewpoints, the same object can appear dramatically in two images. The problem of digits/traffic sign classification is pretty much solved with recent advances (e.g., ladder networks, spatial transformer networks, etc.). What's more, as far as I know, NO paper in the domain adaptation literature RECENTLY published in top conferences tested on digits/traffic signs ALONE. These data might good to generate some plots to illustrate the idea. What's more, Office is NOT the only benchmark real images dataset of 3D objects. I STRONGLY suggested the authors to test it on standard benchmarks. I do not think the authors' claim of the limitation of Office is a solid response. There are several much larger standard benchmarks beyond Office, e.g., 'A Testbed for Cross-Dataset Analysis', 'ImageCLEF Domain Adaptation Dataset'. The other thing is that the authors REIMPLEMENTed the other baseline methods despite the fact that most of them have released the source code. The reimplementation is also different from the original one (e.g., gaussian kernel vs. RBF kernel). What's more, some baseline methods do NOT have hyper-parameter (e.g., [26]), how do you tune it on the validation set? How do I know whether the comparison is fair or not? For pixel-wise reconstruction, due to the much larger space and variance of real images of 3D objects, I do NOT think it is going to work. A reasonable analogy would be autoencoder/decoder. The autoencoder/decoder approaches have been shown to be performed well on MNIST digits. However, as far as I know, no one showed pixel-wise autoencoder/decoder work on real images of 3D objects (e.g., ImageNet). For deep methods on real images of 3D objects, the features need to be much more invariant. These invariances make pixel-wise reconstruction impossible. I do not think one can have both at the same time. For the Domain Separation, the authors claim that 'the shared encoder represents the object and the private encoder represents the background'. Why can not I use some existing segmentation methods to separate the object from the background? It seems much more easier to leverage the existing segmentation methods as the orthogonality constraint might not work for deep networks (see below). For the orthogonality constraint, I think it is NOT going to work in modern deep frameworks. From LeNet to AlexNet is not just scaling up. There are several techniques that contribute a lot. For example, ReLU, Dropout. I really do not think one can train a deep network without all the techniques. For ReLU, it changes all the negative values to zero. For Dropout, the training and testing phases are quite different. During training, randomly zeroing out some (e.g., 50%) activations. For testing, a weight (e.g., 0.5) is put on ALL activations. In terms of distributions of activations, the training and test are probabilistically the same. However, in terms of orthogonality, they are different. Roughly speaking, there are MORE (since ReLU changes the negative parts to zero) than 50% data are zero after ReLU and Dropout if dropout rate is 0.5. To constrain two nonnegative metrics to be orthogonality, at least one of the two corresponding entries need to be zero. This might happy during training since more than 50% data are zero. However, it should be VERY UNSTABLE since for different iteration Dropout will zero out DIFFERENT dimensions. However, during testing as a weight (e.g., 0.5) is put on every dimension rather than zeroing out 50%, most of the data will be positive. The zeros can only be from ReLU not Dropout. For two nonnegative matrices with ~90% positive data, I think it is VERY UNLIKELY to be orthogonal. In summary, even though the writing is excellent, I do NOT think the CURRENT draft is suitable for publication at NIPS. I recommend for a major revision

Confidence in this Review

3-Expert (read the paper in detail, know the area, quite certain of my opinion)


Reviewer 4

Summary

This paper proposes to learn image representations that are partitioned into two subspaces: one is private to each domain and the other is shared across domains. The proposed model is trained jointly by the task in the source domain and image reconstruction from both domains using the partitioned representation. Experiments on various source - target domain datasets show the proposed model outperforms other state-of-the-art unsupervised domain adaptation approaches.

Qualitative Assessment

1. The paper tries to solve an important and challenging problem, i.e., learning representations that are domain–invariant in scenarios where the data distributions during training and testing are different. 2. The idea is interesting and reasonable. It explicitly and jointly models both private and shared components of the domain representations. 3. Extensive experiments are conducted to demonstrate the superiority of DSN. 4. The paper is well written and is easy to follow in most sections.

Confidence in this Review

2-Confident (read it all; understood it all reasonably well)


Reviewer 5

Summary

This paper uses private-shared component analysis to learn private domain specific parameters and shared domain agnostic parameters. The argument is that this will improve domain invariant modeling and hence domain generalization.

Qualitative Assessment

The paper is well written and technically correct. The idea of learning factorized representations is intuitive and appealing. The algorithm offers decent performance benefits on the digit datasets. There are some minor details with the paper. - The authors should mention Daume's "frustratingly easy domain adaptation" paper as this also used factorized representations, though not within a deep network. - There is a promise in the introduction of an "interpretably advantage" that was not proved in the experiments. The authors should comment on this - What does fc3 represent in the text? Which layer of AlexNet is this? - Which experiment supports the claim that the shared representation is more generalizable? - Authors argue that the Office dataset is not a good adaptation dataset, however it is a standard evaluation setup within the adaptation community so without that comparison it is difficult to place this algorithm is proper context.

Confidence in this Review

2-Confident (read it all; understood it all reasonably well)


Reviewer 6

Summary

This paper presents a method to jointly learn a task on a dataset A, and transfer this learning onto another data domain (dataset B). This is done by separating the representation into two parts: a shared part, and a private part. The shared part is extracted by the same network regardless of whether the data comes from A or B, and is trained to represent common features between A and B. The private part is extracted using a different network for each domain. A reconstruction loss is used (scale-invaliant MSE) to enforce that the input can be reconstructed from the sum of the two parts, and a task-specific loss is used on the shared part of the data from the domain A. At the same time, two extra losses are used: one to ensure the shared parts have the same statistic on A and B (either an adversarial loss (DANN), of a Maximum Mean Discrepancy (MMD)), and another one to force the shared and private part to be orthogonal from each other. Results are presented on several pairs of datasets and compared to other methods, as well as two baselines (a lower and upper bounds). The DANN seems to always outperform every other methods. Few reconstruction results are shown, only on two datasets.

Qualitative Assessment

The method presented seems to make sense. It tackles a very useful problem, which is bound to become more and more useful as we produce robots, self driving cars, etc. so we can train them on synthetic data and run them in the real world. The five datasets presented, however, have lots of similarities (mostly digits), and are mostly toy. Therefore, this article seem like a proof of concept of the method, a larger scale dataset (imagenet?) would make the paper much stronger. In particular, the encoders are 2 layers NN (convolutional -> fully connected), which seem very small for large scale data. Also, the method has at least three important hyperparameters (alpha, beta, gamma) in addition to the usual ones (learning rate, decay, regularization, etc.), and it would be good to explain whether these parameters are easy to tune, and how to tune them (as pointed in the paper, one cannot really use labels on dataset B to validate, since these labels are not available). Finally, the figure 2 would benefit from showing more samples, right now it only shows two datasets, and it's hard to understand the figure 2-d .

Confidence in this Review

2-Confident (read it all; understood it all reasonably well)